# Beyond a Spray: Pesticide Application Management in Rural China Based on Quadrilateral Evolutionary Game

**DOI:** 10.3390/ijerph191912096

**Published:** 2022-09-24

**Authors:** Zilu Zhao, Bo Li

**Affiliations:** Business School, Wenzhou University, Wenzhou 325035, China

**Keywords:** pesticide application management, pesticide overuse, retailers, evolutionary game, rural China

## Abstract

Organic pesticides (OP) produce very little environmental contamination compared with conventional pesticides, but their use is low in rural China. Interest conflicts among participants are analysed for the first time to improve pesticide application management (PAM). Retailers, whose roles are usually little-mentioned, were found to be irreplaceable and so were included in the model as players. A quadrilateral evolutionary game is constructed for PAM and used data from field research in servey representative areas in rural China to estimate the future situation. It demonstrates that OP cannot be chosen by most farmers under the current policy and market environment. The simulation showed that: (i) The probability and economic loss of retailers when providing high-concentration pesticide recommendations positively impact OP application. (ii) The organic certification cost and the successful application probability both benefit environmental recovery in the short term, while the advantages outweigh the disadvantages in the long run. (iii) The cost of strict regulations negatively correlate with OP application; while the purchase price and the corresponding premium provided by intermediates positively/negatively correlates with OP application. This suggests that the environment would be better protected by increasing farmers’ pesticide knowledge, reducing retailers’ monopoly of influence, and providing subsidies and guidance for organic certification. Moreover, shortening the food supply chain and reducing regulatory costs would also help.

## 1. Introduction

Pesticides, as one of the production inputs of modern agriculture, occupy an irreplaceable position in resisting biological disasters, avoiding agricultural product reductions and assuring crop productivity. Despite their benefits for crop yield, however, the effects of pesticides are less than desirable when leaving agricultural ecosystems. Depending on the application, fewer than 10% of the applied pesticides reach their targets [1], with the rest (more than 90%) ending up in the environment, contaminating natural resources such as soil, surface water and underground water [2,3]. Some non-target organisms, including bees and other beneficial insects, earthworms, as well as frogs, may be injured or die from pesticides, resulting in a further decline in yields [4,5,6]. Since synthetic pesticides are difficult to degrade, pests with higher resistance survive and evolve, forcing farmers to use more toxic pesticides and creating a vicious circle [7,8]. Numerous studies have shown that people exposed regularly to pesticides are more likely to acquire certain diseases, ranging from skin diseases to cancer [9,10]. New pesticide technologies, which produce less environmental contamination, have become an alternative for pesticide application management (PAM) [11,12,13]. However, due to various reasons, such as the high costs of new technologies, insufficient policy implementation, and high market risks, organic pesticides (OP) cannot completely replace conventional pesticides (CP) in a short period of time [14,15,16].

As the most populous country globally, China is a typical pesticide-dependent agriculture production area. Lack of professional knowledge among farmers and lack of effective certification mechanism has hindered comprehensive PAM efficiency, which is only 35% of that of the US and 40% of that of the UK [17]. Only 12.6% of the surveyed farmers used pesticides at the recommended dosage in the manual, with 73.4% using twice the recommended dosage and 14.0% using more than twice [18]. Hundreds of approaches have been adopted to improve the PAM situation, such as introducing new legislation, establishing regulatory systems and strengthening international cooperation [19]. Unfortunately, multiple yield surveys showed that misuse and abuse of pesticides are still common in rural China [20,21,22]. Many studies found that the farmers’ individual characteristics impact PAM, such as gender, age, education, and risk appetite [23,24,25]. Meanwhile, some external factors, e.g., neighbour choice, official certification, and retailers’ recommendations, also influence PAM significantly [26,27]. In particular, retailers were confirmed as the major information source for farmers’ pesticide application. However, few studies have analysed stakeholders of PAM based on game theory.

Multiple stakeholders, including the governments, consumers, pesticide manufacturers, retailers, non-governmental organisations (NGOs) and the public, impact PAM. It is important to guarantee the balance between benefits and costs of all stakeholders, which is a great challenge for policymakers. OP are less harmful to the environment due to lower toxicity and higher degradability than CP. The transition from CP to OP is the transition from negative externalities to positive ones. Therefore, governments should pay attention to the internalisation of externalities, providing impetus for OP popularisation [28]. Consumers’ feedback information for organic agricultural products (OAPs) is essential for environment-friendly agriculture. The price of OAPs associated with OP greatly affects consumers’ purchasing desires, determining farmers’ production strategies. As the actual users of pesticides, farmers prioritise their own income over environmental health, especially those younger or with larger households [29]. Increased penalties and compliance incentives seem to be ways to encourage farmers to apply OP more frequently. However, due to the large number and small scale of farmers in rural China, it isn’t easy to regulate PAM thoroughly [29].

Retailers, who are paid little attention to by the majority of studies, are scattered throughout rural areas, have close communication and beneficial exchanges with farmers [27]. Compared with policymakers and big manufacturers in remote offices, retailers have a more prominent and sometimes crucial influence on farmers’ pesticide choice. Almost all studies conducted in developing countries suggest the low education levels among rural households. Lack of pesticide knowledge makes it difficult for farmers to apply pesticides precisely. A field study conducted in northern China reveals that the farmers’ average education length is only 6–8 years [30]. A similar distribution of education in rural China can be found in other survey samples [15,31]. By contrast, almost all retailers were literate. More than 90% of retailers had a secondary education, and nearly 65% had college education [32]. However, retailers first care about personal benefits, such as economic income and local reputation, rather than environmental health, and this brings serious consequences [27].

The evolutionary game theory, based on finite rationality, is a game theory combining traditional game theory with Darwin’s theory of biological evolution. Compared with traditional game theory, evolutionary game theory does not require complete rationality or complete information. On the contrary, the game subjects are considered to have finite rationality and to learn in the process of repeated games, to adjust strategies, and maximise their own interests, which is more realistic. At present, evolutionary game theory has been widely used to study the dynamic operation of complex systems involving multiple stakeholders. To achieve sustainable use of cultivated land in China, Xie et al. [33] having conducted a simulation analysis of two evolutionary game models, proposed the necessity of a fallow plan for cultivated land. Xue et al. [28] simulated the interaction process between cooperative network stakeholders in the livestock pollution system. In addition, the evolutionary game model is also applied to complex systems such as air pollution regulation, clean energy development, and dangerous goods logistics management [34,35,36].

Previous research is excellent and provides a rich theoretical basis. However, there are two research gaps worth exploring. (i) Existing research focuses on the deleterious results of different pesticides on the environment, animals or humans in different areas, while conflicts of interest are downplayed. Most studies concentrate on each stakeholder group independently, and the great influence of dynamic interaction strategies on PAM has been ignored. (ii) Due to the complexity of the model, most studies use evolutionary game models with high simplification, selecting only two or three participants for research. Large errors exist between the model and reality. The optimal outcome in a three-party evolutionary game model may not meet the constraints of the improved model when new players are added. For this study, retailers greatly affect farmers’ choice of PAM, ignoring which leads to serious errors in the model.

To fill these gaps, two improvements are implemented in this paper. (i) For the research content, this paper applies the evolutionary game model to PAM for the first time. New solutions are proposed from the game model perspective by analysing stakeholders’ benefits. (ii) For the research method, the traditional evolutionary game model of selecting only two or three game parties is improved upon in this paper. Four stakeholders—namely the governments, retailers, farmers, and consumers—are incorporated into the model. Although the model complexity is doubled, the introduction and analysis of pesticide retailers make the research more realistic, and give stronger and more significant practical guidelines.

## 2. Materials and Methods

### 2.1. Stakeholders and Game Framework

The logical framework of relationships among stakeholders associated with PAM is presented in Figure 1. The central government, local governments, retailers, farmers, and consumers are the core stakeholders of PAM in China, playing the roles of administrator, law-executor, sellers, users, and ultimate undertakers, respectively. To the best of our knowledge, little literature investigates pesticide producers’ influence on PAM. Thus, it is assumed that retailers represent the interests of pesticide producers.

### 2.2. Game Model Assumptions

The following assumptions are proposed.

#### 2.2.1. Assumption 1

Four stakeholders, consisting of local governments, retailers, farmers and consumers, are involved in the evolutionary game model of PAM in rural China. They are all rationally bounded, constantly learning from each other, adjusting their strategic choices, and pursuing the maximisation of their interests through evolution. The probabilities that local governments choose the Strict Regulation and Loose Regulation strategies are g(g∈[0,1]) and 1−g, respectively. The probabilities that retailers choose the Low-Concentration Pesticides (L-CP) and High-Concentration Pesticides (H-CP) recommended strategies are r(r∈[0,1]) and 1−r, respectively. The probabilities that farmers choose the OP and CP strategies are f(f∈[0,1]) and 1−f, respectively. The probabilities that consumers choose the Organic Consumption and Conventional Consumption strategies are c(c∈[0,1]) and 1−c, respectively.

#### 2.2.2. Assumption 2

A higher cost $C_{gh}$ needs to be paid when local governments choose strict regulation. In this case, retailers and farmers need to pay fines when choosing H-CP recommendations or CP applications, defined as $d_{fr}$ and $d_{ff}$, respectively. A lower cost $C_{gl}$ needs to be paid when local governments choose loose regulation. In this case, retailers and farmers can easily avoid regulation with violations going unpunished. Retailers’ significant influence makes farmers apply pesticides at recommended concentrations, including H-CP and L-CP. Farmers can choose the type of pesticides, namely OP or CP. Therefore, four kinds of pesticides are involved in the paper: arranged according to the degree of pollution, there are organic L-CP with the price of $p_{ol}$, organic H-CP with the price of $p_{oh}$, conventional L-CP with the price of $p_{cl}$, and conventional H-CP with the price of $p_{ch}$. Two agricultural products, OAPs and CAPs, are purchased by intermediaries (individuals or businesses) from farmers at the prices of $ap_{o}$ and $ap_{c}$. Note that when the probability of consumers choosing organic consumption is 0, the price of OAPs drop in line with CAPs (i.e., $ap_{o}=ap_{c}$). All parameters in this paragraph, excluding regulatory costs, are transferred from one stakeholder to another in monetary form.

#### 2.2.3. Assumption 3

It is important to know that the behaviour of applying OP alone cannot bring obvious additional benefits to farmers unless they obtain organic certification issued by local governments. However, organic certification imposes tough requirements on farmers’ knowledge level, production technology, and the seeds and seedlings used. The application of organic fertilisers and pesticides, as well as fixed input such as facility renovation, leads to an increased cost for farmers. Moreover, farmers also need to increase their learning costs, which most farmers reject. We use $C_{fo}$ to express the sum of increased costs for organic certification in total, and note that farmers’ attempts to apply for organic certification are not always successful. Natural factors such as soil and water pollution, as well as human factors such as farmers ignoring a certain index or policy changes, may lead to failure. The probability that a farmer successfully applies for organic certification is *θ*. Finally, due to increased labour, the yield is reduced by σ times, whether the application for certification is successful or not.

#### 2.2.4. Assumption 4

When retailers recommend H-CP, an economic loss $E_{r}$ with the probability of β may exist due to credit decline, which depends on farmers’ pesticide knowledge and independent judgment. After purchasing agricultural products, intermediaries who want more profit increase the price of OAPs and CAPs when supplying those to consumers. In our model, the premium is of $\eta_{o}$ times for OAPs and $\eta_{c}$ times for CAPs, respectively. Consumers who purchase OAPs receive the health benefits $H_{tc}$, regardless of whether they choose organic consumption or not; while consumers who choose organic consumption receive the psychological benefits $P_{sc}$, regardless of whether they purchase OAPs or not. The governments gain benefits from this process, notably increased labour efficiency and more investment ($H_{tg}$) or reduced promotion costs ($P_{sg}$). Due to a better environment, consumers and the governments get external benefits ($E_{mij}(i=g,c|j=o,l)$), which represent the environmental improvement brought by popularising OP and L-CP recommendations, respectively. For better differentiation, $E_{mij}$ is superimposable and separable. Note that all parameters in this paragraph are externalities, that is, benefit transformation from one stakeholder to another in monetary form does not exist.

For the convenience of analysis, the symbols and notes of parameters involved in the game analysis are shown in Table 1. 

### 2.3. Game Model Establishment

Based on the assumptions above, the payoff matrices of local governments, retailers, farmers and consumers are shown in Table 2 and Table 3.

According to the payoff matrices shown in Table 2 and Table 3, the dynamic replication equations of the four participants can be derived. To facilitate understanding of the main points of this paper, detailed analysis and calculations are provided in Appendix A. The replicator dynamics equations for local governments, retailers, farmers and consumers can be expressed as follows.
(1)$$\begin{array}{l} F_{\left(g\right)}=\frac{dg}{dt}=g(1-g)(E_{g1}-E_{g2})\\ =g(1-g)[-(C_{gh}-C_{gl})+(1-r)\cdot d_{fr}+(1-f)\cdot d_{ff}] \end{array}$$
(2)$$\begin{array}{l} F_{\left(r\right)}=\frac{dr}{dt}=r(1-r)(E_{r1}-E_{r2})\\ =r(1-r)[-f\cdot (p_{oh}-p_{ol})-(1-f)\cdot (p_{ch}-p_{cl})+g\cdot d_{fr}+\beta \cdot E_{r}] \end{array}$$
(3)$$\begin{array}{l} F_{\left(f\right)}=\frac{df}{dt}=f(1-f)(E_{f1}-E_{f2})\\ =f(1-f)\{c\cdot [\theta \cdot \sigma \cdot ap_{o}+(1-\theta )\cdot \sigma \cdot ap_{c}-ap_{c}]+(1-c)[\sigma \cdot ap_{c}-ap_{c}]\\ -r\cdot (p_{ol}-p_{cl})-(1-r)\cdot (p_{oh}-p_{ch})+g\cdot d_{ff}-C_{fo}\} \end{array}$$
(4)$$F_{\left(c\right)}=\frac{dc}{dt}=c(1-c)(E_{c1}-E_{c2})=c(1-c)[-\theta \cdot f\cdot (\eta_{o}\cdot ap_{o}-\eta_{c}\cdot ap_{c})+P_{sc}]$$

### 2.4. Equilibrium Solution of the Quadrilateral Evolutionary Game

When the four parties’ replicator dynamics equations are equal to 0, the evolutionary game equilibrium is solved as follows:(5)$$\{\begin{array}{c} F_{\left(g\right)}=\frac{dg}{dt}=g(1-g)(E_{g1}-E_{g2})=g(1-g)[-(C_{gh}-C_{gl})+(1-r)\cdot d_{fr}+(1-f)\cdot d_{ff}]=0\\ F_{\left(r\right)}=\frac{dr}{dt}=r(1-r)(E_{r1}-E_{r2})=r(1-r)[-f\cdot (p_{oh}-p_{ol})-(1-f)\cdot (p_{ch}-p_{cl})+g\cdot d_{fr}+\beta \cdot E_{r}]=0\\ \begin{array}{l} F_{\left(f\right)}=\frac{df}{dt}=f(1-f)(E_{f1}-E_{f2})=f(1-f)\{c\cdot [\theta \cdot \sigma \cdot ap_{o}+(1-\theta )\cdot \sigma \cdot ap_{c}-ap_{c}]+(1-c)[\sigma \cdot ap_{c}-ap_{c}]\\ -r\cdot (p_{ol}-p_{cl})-(1-r)\cdot (p_{oh}-p_{ch})+g\cdot d_{ff}-C_{fo}\}=0 \end{array}\\ F_{\left(c\right)}=\frac{dc}{dt}=c(1-c)(E_{c1}-E_{c2})=c(1-c)[-\theta \cdot f\cdot (\eta_{o}\cdot ap_{o}-\eta_{c}\cdot ap_{c})+P_{sc}]=0 \end{array}$$

All equilibrium points can be identified by solving Equation (5). Sixteen special equilibrium points are presented in Equation (5), including $E_{1}(0,0,0,0)$, $E_{2}(0,0,0,1)$, $E_{3}(0,0,1,0)$, $E_{4}(0,0,1,1)$, $E_{5}(0,1,0,0)$, $E_{6}(0,1,0,1)$, $E_{7}(0,1,1,0)$, $E_{8}(0,1,1,1)$, $E_{9}(1,0,0,0)$, $E_{10}(1,0,0,1)$, $E_{11}(1,0,1,0)$, $E_{12}(1,0,1,1)$, $E_{13}(1,1,0,0)$, $E_{14}(1,1,0,1)$, $E_{15}(1,1,1,0)$, $E_{16}(1,1,1,1)$. The area enclosed by these equilibrium points is the equilibrium solution domain of the quadrilateral evolutionary game Ω={(g,r,f,c)|g∈(0,1),r∈(0,1),f∈(0,1),c∈(0,1)}. Additionally, there may be an equilibrium point $E_{17}$ of the hybrid strategy in the system when the conditions in Equation (6) are met:(6)$$\{\begin{array}{l} -(C_{gh}-C_{gl})+(1-r)\cdot d_{fr}+(1-f)\cdot d_{ff}=0\\ -f\cdot (p_{oh}-p_{ol})-(1-f)\cdot (p_{ch}-p_{cl})+g\cdot d_{fr}+\beta \cdot E_{r}=0\\ \begin{array}{l} c\cdot [\theta \cdot \sigma \cdot ap_{o}+(1-\theta )\cdot \sigma \cdot ap_{c}-ap_{c}]+(1-c)[\sigma \cdot ap_{c}-ap_{c}]-r\cdot (p_{ol}-p_{cl})\\ -(1-r)\cdot (p_{oh}-p_{ch})+g\cdot d_{ff}-C_{fo}=0 \end{array}\\ -\theta \cdot \sigma \cdot f\cdot (\eta_{o}\cdot ap_{o}-\eta_{c}\cdot ap_{c})+P_{sc}=0 \end{array}$$

However, $E_{17}$ should be discarded when $E_{17}\notin \Omega$. The strategy combination is asymptotically stable in the dynamic replication system of the multi-party evolutionary game when and only when it is a pure strategy Nash equilibrium. In this model, the asymptotically stable equilibrium point must be the evolutionary stable strategy (ESS) because the equilibrium in this model is a hybrid-strategy Nash equilibrium, and $E_{17}$ is not an ESS. Therefore, this paper does not discuss the asymptotic stability of point $E_{17}$ any further, and only considers the other sixteen equilibrium points.

The local equilibrium points of the Jacobian matrix can usually be used to judge the system stability of evolutionary game models. Friedman [37] proposed a method based on the eigenvalues of the Jacobian matrix to evaluate the stability of the equilibrium point of the system. The signs of all eigenvalues of the Jacobian matrix (i.e., λ) are determined according to the stability theorem of the differential equation [38]. The equilibrium point is unstable if at least one eigenvalue is greater than zero (i.e., λ>0), and the equilibrium point is a sink point if all eigenvalues are less than zero (i.e., λ<0). The calculation and analysis of the Jacobian matrix of sixteen equilibrium points are shown in Appendix B.

The Jacobian matrix of the quadrilateral evolutionary game can be described as follows:(7)$$\begin{array}{l} J=\left[\begin{array}{cccc} \frac{\partial F(g)}{\partial g} & \frac{\partial F(g)}{\partial r} & \frac{\partial F(g)}{\partial f} & \frac{\partial F(g)}{\partial c}\\ \frac{\partial F(r)}{\partial g} & \frac{\partial F(r)}{\partial r} & \frac{\partial F(r)}{\partial f} & \frac{\partial F(r)}{\partial c}\\ \frac{\partial F(f)}{\partial g} & \frac{\partial F(f)}{\partial r} & \frac{\partial F(f)}{\partial f} & \frac{\partial F(f)}{\partial c}\\ \frac{\partial F(c)}{\partial g} & \frac{\partial F(c)}{\partial r} & \frac{\partial F(c)}{\partial f} & \frac{\partial F(c)}{\partial c} \end{array}\right]\\ =\left[\begin{array}{cccc} (1-2g)\left[\begin{array}{c} -(C_{gh}-C_{gl})\\ (1-r)d_{fr}\\ (1-f)d_{ff} \end{array}\right] & g(1-g)(-d_{fr}) & g(1-g)(-d_{ff}) & 0\\ r(1-r)\cdot d_{fr} & (1-2r)\left[\begin{array}{c} -f(p_{oh}-p_{ol})\\ -(1-f)(p_{ch}-p_{cl})\\ g\cdot d_{fr}\\ \beta \cdot E_{r} \end{array}\right] & r(1-r)\left[\begin{array}{c} -(p_{oh}-p_{ol})\\ p_{ch}-p_{cl} \end{array}\right] & 0\\ f(1-f)\cdot d_{ff} & f(1-f)\left[\begin{array}{c} -(p_{ol}-p_{cl})\\ p_{oh}-p_{ch} \end{array}\right] & (1-2f)\left[\begin{array}{c} c[\theta \cdot \sigma \cdot ap_{o}+(1-\theta )\cdot \sigma \cdot ap_{c}-ap_{c}]\\ \left(1-c)[\sigma \cdot ap_{c}-ap_{c}\right]\\ -r(p_{ol}-p_{cl})\\ -(1-r)(p_{oh}-p_{ch})\\ g\cdot d_{ff}-C_{fo} \end{array}\right] & f(1-f)\left[\begin{array}{c} \theta \cdot \sigma \cdot ap_{o}+(1-\theta )\cdot \sigma \cdot ap_{c}-ap_{c}\\ -[\sigma \cdot ap_{c}-ap_{c}] \end{array}\right]\\ 0 & 0 & c(1-c)\left[-\theta \cdot (\eta_{o}\cdot ap_{o}-\eta_{c}\cdot ap_{c})\right] & (1-2c)\left[\begin{array}{c} -\theta \cdot f\cdot (\eta_{o}\cdot ap_{o}-\eta_{c}\cdot ap_{c})\\ P_{sc} \end{array}\right] \end{array}\right] \end{array}$$

#### 2.4.1. Stability Analysis of Equilibrium Points When Local Governments Choose Loose Regulation (1−g)

When local governments choose loose regulation, this means the precondition
(8)$$-(C_{gh}-C_{gl})+(1-s)\cdot d_{fs}+(1-f)\cdot d_{ff}<0$$
is satisfied. The asymptotic stability analysis of the equilibrium points of the dynamic replication system is shown in Table 4.

Two strategy combinations, $E_{2}(0,0,0,1)$ and $E_{4}(0,0,1,1)$, may exist as ESS. $E_{2}(0,0,0,1)$ represents the scenario when retailers choose H-CP recommendations, farmers choose to apply CP, and consumers choose organic consumption. This scenario exists when the stable condition $\theta \cdot (ap_{o}-ap_{c})<(p_{oh}-p_{ch})+C_{fo}$ is met. The left side of the formula refers to the purchase price difference for farmers; and the right side of the formula to the sum of the price difference between organic H-CP and conventional H-CP plus the cost of organic certification. Both sides consider the probability of organic certification. In the scenario in Equation (8), the supply of OAPs is insufficient to satisfy all consumers. However, the strategy change of farmers producing OAPs leads to higher costs, which cannot be compensated for by increased profits. The illegal behaviour of applying CP and H-CP recommendations are not fined because of local governments’ loose regulation. Thus, farmers’ strategies evolve after multiple games into giving up the application of OP, and applying CP instead to ensure their income. Similarly, two preconditions should be met for $E_{4}(0,0,1,1)$.

However, both scenarios with the least environmental contamination, $E_{7}(0,1,1,0)$ and $E_{8}(0,1,1,1)$, fail to become possible ESS. The retailers’ strategies are unstable in both scenarios, which means that strict regulation by local governments is necessary to achieve better PAM. Otherwise, retailers do not choose to sacrifice their own interests for environmental improvement, which is consistent with the previous analysis.

#### 2.4.2. Stability Analysis of Equilibrium Points When Local Governments Choose Strict Regulation (g)

When local governments choose strict regulation, this means the precondition
(9)$$-(C_{gh}-C_{gl})+(1-s)\cdot d_{fs}+(1-f)\cdot d_{ff}>0$$
is satisfied. The asymptotic stability analysis of the equilibrium points of the dynamic replication system is shown in Table 5.

Four strategy combinations, $E_{10}(1,0,0,1)$, $E_{11}(1,0,1,0)$, $E_{12}(1,0,1,1)$ and $E_{14}(1,1,0,1)$, may exist as ESS when local governments choose strict regulation. Different requirements need to be met in different scenarios, which are not described here for reasons of paper length. However, two scenarios with the least environmental contamination, $E_{15}(1,1,1,0)$ and $E_{16}(1,1,1,1)$, fail to become possible ESS. In these two scenarios, the environmental protection strategies of using retailers’ L-CP recommendations and farmers’ OP applications were chosen and evolved to be stable. However, local governments’ strict regulatory strategy was unstable because the good behaviour of retailers and farmers made regulation unnecessary. Local governments then spend their budgets elsewhere than PAM. The looser regulation leads retailers to choose H-CP recommendations again and the situation degenerates into a sub-optimal one. How to reduce PAM contamination and ensure its stability deserves further study.

### 2.5. Data Sources

Based on the theoretical analysis above, six scenarios, $E_{2}(0,0,0,1)$, $E_{4}(0,0,1,1)$
$E_{10}(1,0,0,1)$, $E_{11}(1,0,1,0)$, $E_{12}(1,0,1,1)$ and $E_{14}(1,1,0,1)$, could eventually become the ESS when corresponding requirements are met. However, none of these is an optimal scenario. This study aims to increase the use of L-CP and OP in agricultural production, which are not in conflict and can be implemented simultaneously. Therefore, four scenarios, $E_{7}(0,1,1,0)$, $E_{8}(0,1,1,1)$ $E_{15}(1,1,1,0)$ and $E_{16}(1,1,1,1)$ meet the ideal requirements as qualified strategy combinations. Unfortunately, none exist as ESS. How to prolong the duration of these four scenarios in the evolutionary game process is the focus of the following discussion.

At present, a wide variety of pesticides are applied in China. Pesticide requirements for different crops vary greatly. Factors such as climatic conditions, geographical location, and past application all greatly impact PAM. There are also differences in pesticides from different manufacturers. The officially published lists are chosen to distinguish OP from CP. Moreover, economic crops, facility vegetables and particularly fruits have a larger profit, leading to much higher pesticide application than food crops.

To make the simulation more realistic, a field survey combined with an online questionnaire was undertaken in Apr. 2021 for the latest data. Servey representative areas were selected in Henan, one of the major agricultural provinces in China. The surveyed areas are shown in Figure 2. Purple cabbage as a significant contributor to meeting residents’ diets is the objective of this study. The 237 surveyed households own an average of 3.72 mu (2.48 ha) of facility land, taking the family as the basic unit. The weighted average shows an average of 8000 jin (4000 kg) of facility vegetables produced on each mu (0.67 ha).

The survey found that CP was applied most frequently (87.3%), with an average cost of 300 CNY/mu (68.85 USD/ha) as initially recommended in the manual. (The average exchange rate is USD 1 = CNY 6.52 in April 2021.) However, the recommended concentration by retailers then increased to 400 CNY/mu (91.80 USD/ha). Meanwhile, OP was applied at a lower rate (12.7%), with an average cost of 1500 CNY/mu (344.25 USD/ha), whereas retailers typically recommend about 2000 CNY/mu (459.00 USD/ha).

Most of the agricultural products sold in this survey are CAPs. Considering the high fluctuation in vegetable prices, as well as the price differences between different varieties and weights, the weighted average according to the area of the interviewed farmers was taken for calculations. The price of 0.75 CNY/jin (0.23 USD/kg) was used as the purchase price for conventional purple cabbages, and 5 CNY/jin (1.53 USD/kg) for organic ones. According to several farmers who had successfully applied for organic certification, yield reduction is set at 25%. Also, at least CNY 20,000 (USD 3067.48) are needed for the organic certification. A large price difference appears when researching the price of purple cabbages in supermarkets and agricultural wholesale markets. Various factors affect the price of agricultural products, such as the distance between place of production and sale, urban consumption level, warehouse management level, and even the location of goods. We assumed the price of conventional purple cabbages on sale increased 5 times to 3.75 CNY/jin (1.15 USD/kg), and that of organic purple cabbages increased 8 times to 40 CNY/jin (12.27 USD/kg). In addition, several officials from the local agriculture department were interviewed to get a more accurate regulation cost. Strict supervision costs are estimated at 2000 CNY/mu (460.12 USD/ha) for the semester, and are halved for loose regulation. The fine for retailers is CNY 500 (USD 76.69) each time, and that for farmers is CNY 1500 (USD 230.06). Moreover, we assume that when retailers are questioned, there is a 5% probability that they would make H-CP recommendations, resulting in a credit decline and economic loss of CNY 500 (USD 76.69) each time, whether OP or CP is sold. Finally, in an online survey of consumers, the median psychological expectation of paying for health benefits was CNY 18 (USD 2.76) each day. Research shows that Chinese residents consume about 200 jin (100 kg) of vegetables per person per year, which means the price of CAPs has an average upward space of CNY 36 (USD 5.32) compared with OAPs.

In addition to 237 practitioners, 13 pesticide retailers, 5 local agricultural officials and 7 experts who engaged in agricultural research were interviewed to make the initial probabilities more realistic. The initial probabilities are set as follows. For local governments g=0.5, which means the local governments have an average 50% probability of choosing strict regulation. Similarly, for retailers, r=0.1, for farmers f=0.2, and for consumers c=0.8. Considering the complexity of PAM in China, numerical simulation can intuitively reveal the dynamic evolution process of stakeholders, which is crucial for providing appropriate policy suggestions. The following numerical simulations were performed with MATLAB 2021a. The evolution process is shown in Figure 3.

## 3. Result

The stakeholders are identified as follows: consumers, retailers, farmers, and local governments. Firstly, the red line indicates that consumers quickly choose the organic consumption strategy and keep it unchanged for a long time. The only change occurs when farmers choose to apply OP with a higher probability. Consumers can obtain the same benefits with less payment by choosing conventional consumption due to lots of OAPs on sale. It can be calculated that *f* = 0.8668, that is, consumers change strategies when the probability of farmers choosing OP application is greater than 86.68%. Note that consumers’ conventional consumption choices greatly reduce farmers’ benefits when choosing to apply OP. This value is so large that it looks like a mutation on the image. When the probability of farmers choosing to apply OP drops sharply, consumers change their strategy back to organic consumption again. Correspondingly, retailers and local governments both have short-term fluctuations due to changes in farmers’ behaviour.

Secondly, the blue line indicates that the probability of retailers choosing L-CP recommendations fluctuates with the probability of strict regulations. When local governments choose loose regulations with a high probability, retailers are more likely to go unpunished for un-environmental behaviours. At this point, the H-CP recommendations are more in the retailers’ interest. Retailers do not spontaneously choose L-CP recommendations since the probability of economic losses from fines is bearable. When the probability of strict regulation increases, retailers are more likely to opt for L-CP recommendations, resulting from the fines outweighing the possible benefits. In turn, the increased probability of L-CP recommendations means that local governments are less likely to receive fines from retailers. When the fines are not enough to compensate for the increased cost of strict regulation, local governments choose to relax regulation again, leading retailers to increase the probability of H-CP recommendations. Retailers stabilize in L-CP recommendations after multiple evolutions.

Thirdly, the yellow line indicates that the probability of farmers choosing OP fluctuates with the probability of strict regulation. Facing the price difference between OP and CP, farmers integrate organic certification costs and possible benefits. At present, farmers cannot get enough income from the high profits of OAPs due to long supply chains. Due to the high cost of organic certification, farmers are less willing to produce OAPs. Thus, organic agriculture and pesticides are not selected by most farmers, which is consistent with our survey and other studies. However, farmers have to choose to apply for organic certification unless they bear higher losses when facing strict regulations. Hence, farmers temporarily increase the probability of applying OP, which also means that the probability of local governments collecting fines decreases continuously. As long as the fine cannot bridge the gap between strict regulation and loose regulation, local governments change strategies to loosen regulation, leading farmers to choose CP again. Unlike retailers, farmers do not choose the environmentally friendly strategy after multiple games. After nearly half-hundred data simulations with different parameters and models, it is confirmed that this situation could not be explained by a single variable or a single numerical comparison.

Finally, the blue line indicates that the probability of local governments choosing strict regulation fluctuates with other stakeholders’ strategy choices. Since consumers quickly choose organic consumption and remain unchanged, the indirect impact on local governments’ strategies can be ignored. The smaller the probabilities of retailers and farmers are, the higher the fines collected by local governments when choosing strict regulation. Because the fines collected from farmers cannot make up for the increased cost of strict supervision, local governments tend to reduce strict supervision compared to the initial position. The game between local governments and farmers repeats continually after retailers’ strategies evolve to ESS, forming a periodic stable game.

## 4. Discussion

Since tiny changes in key parameters can lead to great changes in the strategies of stakeholders, the analysis of these key parameters is crucial, helping infer stakeholders’ strategy choices in different scenarios. This study analyses the following key parameters: the probability (β) of economic loss ($E_{r}$) of retailers when providing H-CP recommendations, the organic certification cost ($C_{fo}$) and the successful probability (θ) for farmers, the strict regulation cost for local governments ($C_{gh}$), as well as the purchase price of OAPs ($ap_{o}$) and the corresponding premium ($\eta_{o}$) from farmers. The strategic choices of farmers and retailers are our focus due to their remarkable impact on the environment. In the following study, only the parameter under study is changed each time, with other parameters unchanged.

### 4.1. The Probability (β) and Economic Loss ($E_{r}$) of Retailers

We set β = 5%, 20%, 50%, 0%, and corresponding $E_{r}$ = 500, 800, 1200, 0 CNY/mu ha (115.03, 183.60, 276.07, 0 USD/ha) to analyse the influence of probability and economic losses of retailers due to H-CP recommendations on stakeholders’ strategies. The simulation results of the four players are shown in Figure 4. It can be seen from Figure 4b that when those two parameters increase, the time when retailers stabilise at L-CP recommendations is reduced from 100 times to 30 times. It can be understood that this change reduces pesticide pollution by about 70% in the first 100 games, due to farmers never choosing OP, and the periodic changes (2.48%) and range changes (4.29%) both being small. When these two parameters continue to increase, as shown in Figure 4c, the retailers choose L-CP recommendations after 10 games and their choice remains unchanged, reducing pesticide pollution by about 90% in the first 100 games. By contrast, as shown in Figure 4d, the probability of choosing the H-CP recommendations increases when farmers completely trust retailers. At present, retailers’ status is close to a knowledge monopoly in rural China, significantly impacting rural households who receive low education. This improvement is small, but the time it takes to evolve to stable L-CP recommendations significantly increases. Breaking the existing knowledge monopoly of retailers in rural areas, especially regarding pesticides, can be an innovative method to reduce H-CP damage to ecosystems.

### 4.2. The Organic Certification Cost ($C_{fo}$) for Farmers

We set $C_{fo}$ = 20,000, 19,000, 18,000, 22,000 CNY/ha (460.12, 437.12, 414.11, 506.13 USD /ha) to analyse the impact of the organic certification cost on stakeholders’ strategy choices. The simulation results of four players are shown in Figure 5. It can be seen from Figure 5b that when the organic certification cost parameter is slightly reduced (5%), the probability of farmers applying OP increases significantly. The time to evolve to the stable periodic game is shortened from about 120 times to less than 10 times. The steady-state period can be considered no significant change (3.85%). The minimum probability of choosing OP increased from 0.04% to 32.6%, and the maximum probability increased from 76.65% to 83.70%, which means that the overall probability of choosing OP increased from 38.39% to 58.11% in the steady-state. However, retailers’ strategies are stabilised at H-CP recommendations, which means greater ecological pollution when farmers choose CP. Such change benefits environmental recovery in the short term, but the advantage exceeds the disadvantage in the long run. When the parameter is reduced by 10%, as shown in Figure 5c, farmers are more likely to choose OP. Although retailers still choose H-CP recommendations for a time, stabilising strategy choices after 5 game times, the overall ecological pollution is greatly reduced. By choosing loose regulation, local governments also had stable strategy choices, with consumers and farmers having a periodic game. By contrast, when the parameter increases, as shown in Figure 5d, farmers do not get more benefits from OAPs compared to CAPs, causing OP application to drop. Since local governments get enough fines to cover the increased costs of strict regulation, retailers evolved to L-CP recommendations for fewer payments, posing a more serious ecological threat than the initial scenario. To sum up, when the parameter increases slightly, it benefits environmental recovery in the short term, while the advantages outweigh the disadvantages in the long run; when greatly increased, however, the parameter results in significant environmental improvement.

In addition, as shown in Figure 6, the same relationship appears when simulating the success probability of organic certification (θ). Here, we only provide the evolutionary process and do not describe it in detail.

### 4.3. The Strict Regulation Cost ($C_{gh}$) of Local Governments

We set $C_{gh}$ = 2000, 1800, 1600, 2400 CNY/mu (460.12, 414.11, 368.10, 552.15 USD/ha) to analyse the impact of different strict regulation costs on stakeholders’ strategies. The simulation results of the four players are shown in Figure 7. It can be seen from Figure 7b that when the parameter decreases slightly (10%), farmers are more likely to choose OP. The evolution period reduces by 10.45%. The minimum probability of choosing CP increases from 0.04% to 23.01%, and the maximum probability increases from 76.65% to 84.11%, which means that the average probability of choosing OP increases from 38.39% to 53.56%. As shown in Figure 7c, when the parameter drops by another 10%, no periodic repeated games arise as before. The probability of farmers choosing OP remains around 82.75%, which could greatly reduce pesticide pollution. Since the pollution effect of organic H-CP is far less than that of conventional L-CP, retailers opted for H-CP recommendations as an acceptable price. The only downside is that farmers lost parts of their income from buying organic H-CP rather than organic L-CP. By contrast, Figure 7d showed a bad situation when strict regulation costs increased by 20%. Neglecting retailers’ options for L-CP recommendations, farmers choose CP for a longer period. The evolution period increases by 166.7%, and the average value of OP application decreases by 35.9%, implying a total decrease of 59.9% after the stabilised evolution. To sum up, reducing strict regulation costs can help reduce the CP application and greatly contribute to strengthening environmental protection.

### 4.4. The Purchase Price of OAPs ($ap_{o}$) and Premium ($\eta_{o}$)

We set $ap_{o}$ = 400,000, 450,000, 600,000, 350,000 CNY/mu (92,024.54, 103,527.61, 138.036.81, 80.521.47 USD/ha), and corresponding $\eta_{o}$ = 8, 7, 5, and 9 times to analyse the impact of different OAP purchase prices on stakeholders’ strategies with different supply chains. The simulation results are shown in Figure 8. Note that the OAP purchase prices do not exceed consumers’ psychological needs among the four sets of parameters displayed. It can be seen from Figure 8b that when the increase in the OAP purchase prices is small (12.50%), farmers have a higher probability of choosing to apply OP. The evolution cycle is greatly shortened to 6.89% of the original. The minimum probability of applying OP increases from 0.04% to 71.50%, and the maximum probability increases from 76.65% to 97.34%, which means that the average probability of applying OP increases from 38.39% to 84.42%. The game between farmers and consumers becomes the focus of this scenario. Retailers quickly choose H-CP recommendations, and local governments choose loose regulation in 10 game times. Similarly, retailers’ H-CP recommendations are an acceptable price to pay. It can be seen from Figure 8c that when the OAP purchase price increases by 50% of the original, farmers choose OP with a higher probability, and the evolution period shortens to 3.86% of the original. No cyclical fluctuations are observed in those two scenarios, the same as the 300-time evolution results. However, farmers’ strategies do not change drastically compared with the previous 10 evolutions. Conversely, the OAP purchase prices decrease compared to the control group if the supply chain becomes more inefficient, as shown in Figure 8d. At this time, farmers’ income from OAPs is less than that of CAPs, and farmers choose to apply CP and ignore the strict regulation, adding further environmental damage with L-CP recommendations from retailers. To sum up, both controlling the length of the supply chain and increasing the purchase price of OAPs could help popularise OP applications, as well as meet consumer demand and increase farmers’ income.

## 5. Conclusions

Based on the background of promoting OP application and reducing environmental pollution, this study has analysed PAM in rural China and constructed, for the first time, a quadrilateral game model for four stakeholders, local governments, retailers, farmers, and consumers. The study shows that six scenarios can be ESS under all sixteen different scenarios, while the optimal state with minimal environmental damage is not stable. To explore the reasons for this, our study estimates the future development trend based on data obtained in some representative areas in Henan, China. The simulation showed that consumers choose organic consumption, retailers choose low-concentration pesticide recommendations after multiple games, and farmers and local governments fall into periodically repeated games.

To prolong the duration of environmentally-friendly choices, sensitivity analyses of some key parameters in the model were performed. Among these, the probability (β) of economic loss ($E_{r}$) of retailers when providing H-CP recommendations had a positive impact on OP application. When increased slightly, both the organic certification cost ($C_{fo}$) and the successful application probability (θ) benefited environmental recovery in the short term, while the advantages outweighed the disadvantages in the long run; when greatly increased, these two parameters resulted in significant environmental improvement. Also, the strict regulation costs ($C_{gh}$) for local governments has a negative correlation with OP application. The OAP purchase price ($ap_{o}$) and the corresponding premium ($\eta_{o}$) provided by intermediates have a positive/negative correlation with OP application.

Based on the analysis above, there is huge room for improvement in pesticide contamination in rural China. The most urgent issue is to increase farmers’ pesticide knowledge so as to reduce the problematic knowledge monopoly held by retailers. Methods such as providing free training and remote online guidance are suggested to decrease retailers’ personal influence. Moreover, it is important to increase subsidies and guidance for organic certification, as well as improve regulatory efficiency. Lastly, optimising supply chain management is another considerable way to increase farmers’ income and indirectly reduce the use of toxic pesticides.

There are still some limitations in the current study. For example, an online questionnaire is used to investigate consumers’ demand for organic agricultural products, which excludes some users who do not have access to using mobile phones, such as the elderly or low-income people. In addition, the real data of our investigation is focused on purple cabbage in one major production area. The data analysed needs to be adjusted accordingly when analysing purple cabbages from other sources, and the model needs to be improved when analysing perishable vegetables such as tomatoes. Therefore, collecting more data and adjusting or improving the model would be another meaningful research effort.

## Figures and Tables

**Figure 1 ijerph-19-12096-f001:**
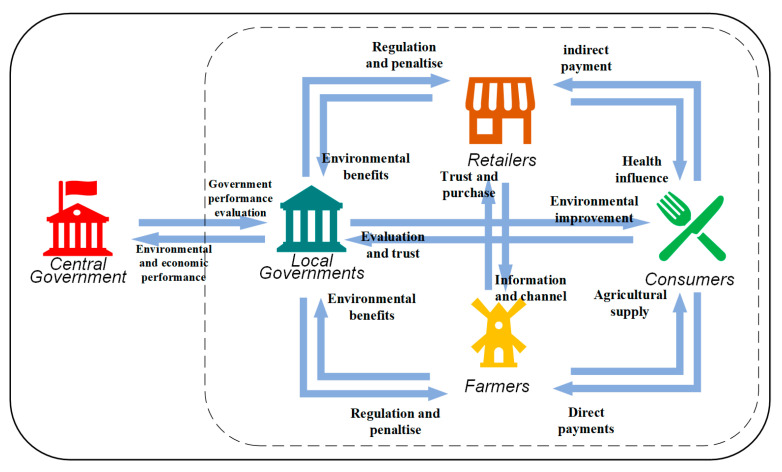
The logical framework of PAM stakeholders’ relationships.

**Figure 2 ijerph-19-12096-f002:**
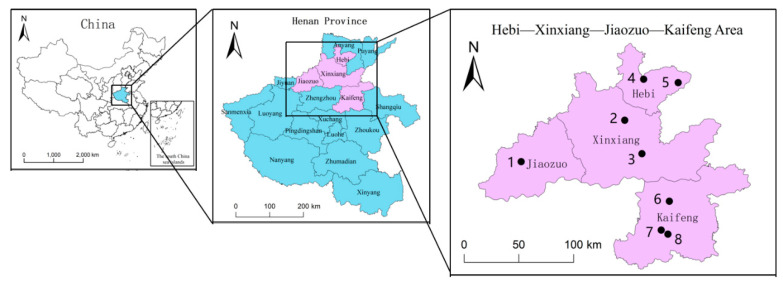
Locations of the field survey area and representative villages.

**Figure 3 ijerph-19-12096-f003:**
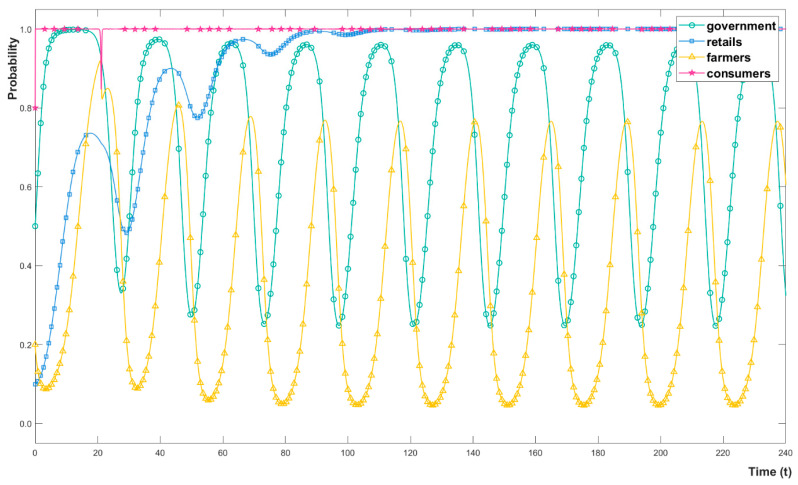
The evolutionary process of stakeholders’ probability of strategy choice.

**Figure 4 ijerph-19-12096-f004:**
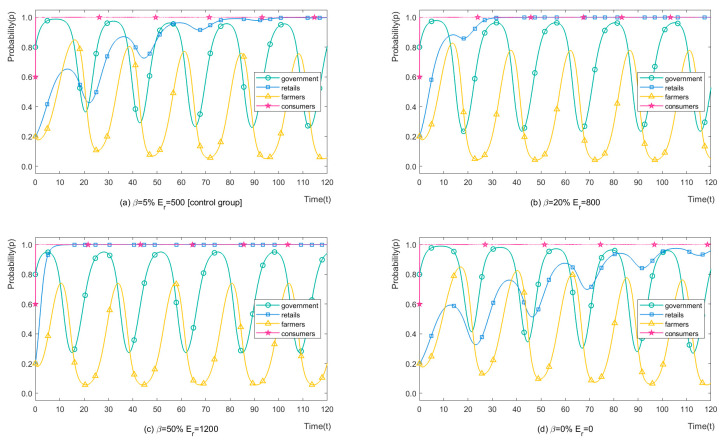
The strategy evolution process of stakeholders with different β and $E_{r}$.

**Figure 5 ijerph-19-12096-f005:**
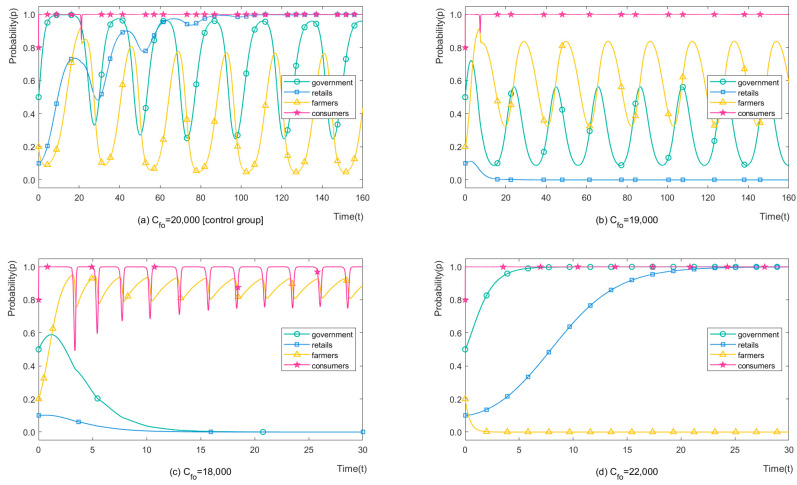
The strategy evolution process of stakeholders with different $C_{fo}$.

**Figure 6 ijerph-19-12096-f006:**
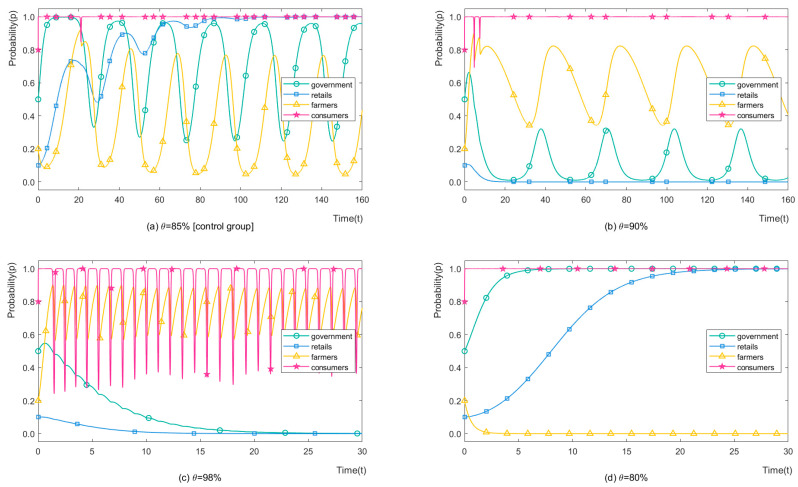
The strategy evolution process of stakeholders with different *θ*.

**Figure 7 ijerph-19-12096-f007:**
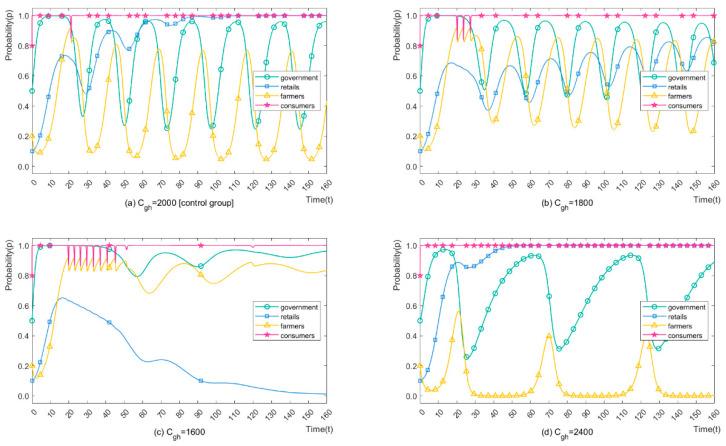
The Strategies Evolution Process of Stakeholders with Different $C_{gh}$.

**Figure 8 ijerph-19-12096-f008:**
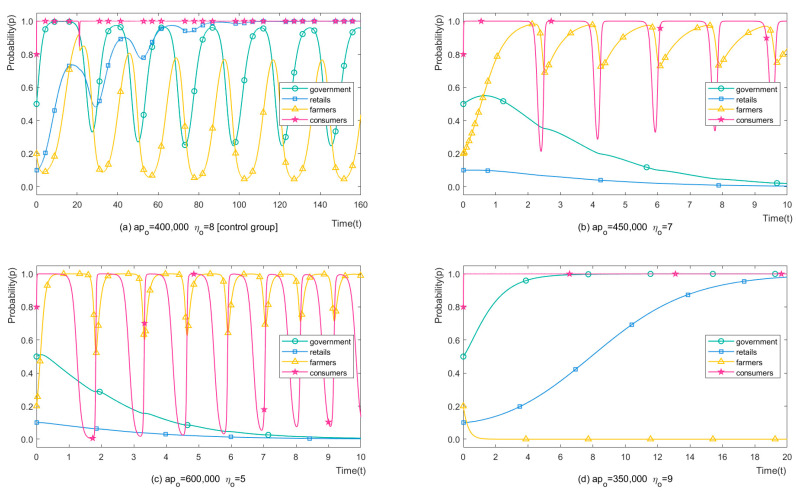
The strategy evolution process of stakeholders with different $ap_{o}$ and $\eta_{o}$.

**Table 1 ijerph-19-12096-t001:** The meaning of parameters and notes about them.

Parameters	Meanings	Notes
$C_{gh}/C_{gl}$	Costs of local governments when choosing strict/loose regulation	$0<C_{gl}<C_{gh}$
$d_{fg}$	Fines levied by local governments	$d_{fg}>0$
$E_{mij}$	External environmental benefits (i=g,c|j=o,l)	$E_{mij}>0$
$H_{ti}$	Health benefits when consumers purchase OAPs (i=g,c)	$H_{ti}>0$
$P_{si}$	Psychological benefits when consumers choose organic consumption (i=g,c)	$P_{si}>0$
$p_{ab}$	Price of different pesticides (a=o,c|b=h,l)	$p_{ab}>0$
$d_{fr}$	Fines of retailers when choosing H-CP recommendations	$d_{fr}>0$
β	Probability of economic loss when choosing H-CP recommendations	0<β<1
$E_{r}$	Economic loss due to credit decline	$E_{r}>0$
$C_{fo}$	Costs of fixed facilities and learning pressure for organic certification	$C_{fo}>0$
θ	Probability of successfully applying for organic certification	0<θ<1
σ	Yield decrease due to increased labour	0<σ<1
$ap_{o}/ap_{c}$	The price of OAPs/CAPs	$0<ap_{c}<ap_{o}$
$\eta_{o}/\eta_{c}$	The price increase multiple of OAPs/CAPs	$1<\eta_{c}<\eta_{o}$
$d_{ff}$	Fines of farmers when choosing CP	$d_{ff}>0$

**Table 2 ijerph-19-12096-t002:** Quadrilateral evolutionary game payoff matrix when local governments choose strict regulation (g).

**Retails**	Farmers			
	Organic Pesticides (f)		Conventional Pesticides (1−f)	
	Consumers		Consumers	
	Organic Consumption (c)	Conventional Consumption (1−c)	Organic Consumption (c)	Conventional Consumption (1−c)
L-CP recommendations (r)	$-C_{gh}+E_{mgo}+E_{mgl}+H_{tg}+P_{sg}$	$-C_{gh}+E_{mgo}+E_{mgl}+H_{tg}$	$\begin{array}{l} -C_{gh}+d_{fg}\\ +E_{mgl}+P_{sg} \end{array}$	$-C_{gh}+d_{fg}+E_{mgl}$
	$p_{ol}$	$p_{ol}$	$p_{cl}$	$p_{cl}$
	$\begin{array}{l} \theta \cdot \sigma \cdot ap_{o}+(1-\theta )\cdot \sigma \cdot ap_{c}\\ -p_{ol}-C_{fo} \end{array}$	$\sigma \cdot ap_{c}-p_{ol}-C_{fo}$	$ap_{c}-p_{cl}-d_{ff}$	$ap_{c}-p_{cl}-d_{ff}$
	$\begin{array}{l} -\theta \cdot \sigma \cdot \eta_{o}\cdot ap_{o}-(1-\theta )\cdot \eta_{c}\cdot ap_{c}\\ +H_{tc}+E_{mco}+E_{mcl}+P_{sc} \end{array}$	$-\eta_{c}\cdot ap_{c}+H_{tc}+E_{mco}+E_{mcl}$	$-\eta_{c}\cdot ap_{c}+E_{mcl}+P_{sc}$	$-\eta_{c}\cdot ap_{c}+E_{mcl}$
H-CP recommendations (1−r)	$-C_{gh}+d_{fg}+E_{mgo}+H_{tg}+P_{sg}$	$-C_{gh}+d_{fg}+E_{mgo}+H_{tg}$	$-C_{gh}+d_{fg}+H_{tg}$	$-C_{gh}+d_{fg}$
	$p_{oh}-d_{fr}-\beta \cdot E_{r}$	$p_{oh}-d_{fr}-\beta \cdot E_{r}$	$p_{ch}-d_{fr}-\beta \cdot E_{r}$	$p_{ch}-d_{fr}-\beta \cdot E_{r}$
	$\begin{array}{l} \theta \cdot ap_{o}+(1-\theta )\cdot \sigma \cdot ap_{c}\\ -p_{oh}-C_{fo} \end{array}$	$\sigma \cdot ap_{c}-p_{oh}-C_{fo}$	$ap_{c}-p_{ch}-d_{ff}$	$ap_{c}-p_{ch}-d_{ff}$
	$\begin{array}{l} -\theta \cdot \eta_{o}\cdot ap_{o}-(1-\theta )\cdot \eta_{c}\cdot ap_{c}\\ +H_{tc}+E_{mco}+P_{sc} \end{array}$	$-\eta_{c}\cdot ap_{c}+H_{tc}+E_{mco}$	$-\eta_{c}\cdot ap_{c}+P_{sc}$	$-\eta_{c}\cdot ap_{c}$

**Table 3 ijerph-19-12096-t003:** Quadrilateral evolutionary game payoff matrix when local governments choose loose regulation (1−g).

**Retails**	Farmers			
	Organic Pesticides (f)		Conventional Pesticides (1−f)	
	Consumers		Consumers	
	Organic Consumption (c)	Conventional Consumption (1−c)	Organic Consumption (c)	Conventional Consumption (1−c)
L-CP recommendations (r)	$-C_{gl}+E_{mgo}+E_{mgl}+H_{tg}+P_{sg}$	$-C_{gl}+E_{mgo}+E_{mgl}+H_{tg}$	$-C_{gl}+E_{mgl}+P_{sg}$	$-C_{gl}+E_{mgl}$
	$p_{ol}$	$p_{ol}$	$p_{cl}$	$p_{cl}$
	$\begin{array}{l} \theta \cdot \sigma \cdot ap_{o}+(1-\theta )\cdot \sigma \cdot ap_{c}\\ -p_{ol}-C_{fo} \end{array}$	$\sigma \cdot ap_{c}-p_{ol}-C_{fo}$	$ap_{c}-p_{cl}$	$ap_{c}-p_{cl}$
	$\begin{array}{l} -\theta \cdot \eta_{o}\cdot ap_{o}-(1-\theta )\cdot \eta_{c}\cdot ap_{c}\\ +H_{tc}+E_{mco}+E_{mcl}+P_{sc} \end{array}$	$-\eta_{c}\cdot ap_{c}+H_{tc}+E_{mco}+E_{mcl}$	$-\eta_{c}\cdot ap_{c}+E_{mcl}+P_{sc}$	$-\eta_{c}\cdot ap_{c}+E_{mcl}$
H-CP recommendations (1−r)	$-C_{gl}+E_{mgo}+H_{tg}+P_{sg}$	$-C_{gl}+E_{mgo}+H_{tg}$	$-C_{gl}+P_{sg}$	$-C_{gl}$
	$p_{oh}-\beta \cdot E_{r}$	$p_{oh}-\beta \cdot E_{r}$	$p_{ch}-\beta \cdot E_{r}$	$p_{ch}-\beta \cdot E_{r}$
	$\begin{array}{l} \theta \cdot ap_{o}+(1-\theta )\cdot \sigma \cdot ap_{c}\\ -p_{oh}-C_{fo} \end{array}$	$\sigma \cdot ap_{c}-p_{oh}-C_{fo}$	$ap_{c}-p_{ch}$	$ap_{c}-p_{ch}$
	$\begin{array}{l} -\theta \cdot \eta_{o}\cdot ap_{o}-(1-\theta )\cdot \eta_{c}\cdot ap_{c}\\ +H_{tc}+E_{mco}+P_{sc} \end{array}$	$-\eta_{c}\cdot ap_{c}+H_{tc}+E_{mco}$	$-\eta_{c}\cdot ap_{c}+P_{sc}$	$-\eta_{c}\cdot ap_{c}$

**Table 4 ijerph-19-12096-t004:** Stability analysis of equilibrium points when local governments choose loose regulation (1−g).

**Equilibrium Points**	$\lambda_{1}\;$	$\lambda_{2}$	$\lambda_{3}$	$\lambda_{4}$	Stability	Stable Conditions
$E_{1}(0,0,0,0)$	−	−	−	+	Unstable	\
$E_{2}(0,0,0,1)$	−	−	U	−	ESS	$\theta \cdot (ap_{o}-ap_{c})<(p_{oh}-p_{ch})+C_{fo}$
$E_{3}(0,0,1,0)$	−	−	+	U	Unstable	\
$E_{4}(0,0,1,1)$	−	−	U	U	ESS	$\left(p_{oh}-p_{ch})+C_{fo}<\theta \cdot \sigma \cdot (ap_{o}-ap_{c}\right)$ $\theta \cdot (\eta_{o}\cdot ap_{o}-\eta_{c}\cdot ap_{c})<P_{sc}$
$E_{5}(0,1,0,0)$	−	+	−	+	Unstable	\
$E_{6}(0,1,0,1)$	−	+	U	−	Unstable	\
$E_{7}(0,1,1,0)$	−	+	+	U	Unstable	\
$E_{8}(0,1,1,1)$	−	+	U	U	Unstable	\

Note: U stands for the uncertainty of the sign.

**Table 5 ijerph-19-12096-t005:** Stability analysis of equilibrium points when local governments choose strict regulation (g).

**Equilibrium Points**	$\lambda_{1}\;$	$\lambda_{2}$	$\lambda_{3}$	$\lambda_{4}$	Stability	Stable Conditions
$E_{9}(1,0,0,0)$	−	U	U	+	Unstable	\
$E_{10}(1,0,0,1)$	−	U	U	−	ESS	$d_{fr}+\beta \cdot E_{r}<p_{ch}-p_{cl}$ $\theta \cdot \sigma \cdot ap_{o}+(1-\theta )\cdot \sigma \cdot ap_{c}-ap_{c}<(p_{oh}-p_{ch})-d_{ff}+C_{fo}$
$E_{11}(1,0,1,0)$	−	U	U	U	ESS	$d_{fr}+\beta \cdot E_{r}<p_{oh}-p_{ol}$ $(p_{oh}-p_{ch})+C_{fo}-d_{ff}<\sigma \cdot ap_{c}-ap_{c}$ $P_{sc}<\theta \cdot (\eta_{o}\cdot ap_{o}-\eta_{c}\cdot ap_{c})$
$E_{12}(1,0,1,1)$	−	U	U	U	ESS	$d_{fr}+\beta \cdot E_{r}<p_{oh}-p_{ol}$ $(p_{oh}-p_{ch})+C_{fo}-d_{ff}<\theta \cdot \sigma \cdot ap_{o}+(1-\theta )\cdot \sigma \cdot ap_{c}-ap_{c}$ $\theta \cdot (\eta_{o}\cdot ap_{o}-\eta_{c}\cdot ap_{c})<P_{sc}$
$E_{13}(1,1,0,0)$	−	U	U	+	Unstable	\
$E_{14}(1,1,0,1)$	−	U	U	−	ESS	$p_{ch}-p_{cl}<d_{fr}+\beta \cdot E_{r}$ $(p_{ol}-p_{cl})+C_{fo}-d_{ff}<\theta \cdot \sigma \cdot ap_{o}+(1-\theta )\cdot \sigma \cdot ap_{c}-ap_{c}$
$E_{15}(1,1,1,0)$	+	U	U	U	Unstable	\
$E_{16}(1,1,1,1)$	+	U	U	U	Unstable	\

## Data Availability

The data that support the findings of this study are available from the corresponding author upon reasonable request.

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
