# Peer review of "Beyond a Spray: Pesticide Application Management in Rural China Based on Quadrilateral Evolutionary Game"

_ijerph, 2022, doi:10.3390/ijerph191912096_

Round 1

Reviewer 1 Report

Dear Authors,

I reviewed the paper, but before any deeper comments and suggestions, the paper should be written in accordance with the Instructions for Authors. 

Editors should contact you regarding next steps.

Author Response

Thank you for your significant reminding.

Please see the attachment。

Reviewer 2 Report

The issue of entering the market of organic plant protection products is a very important in the context of striving to reduce the negative effects of intensive agriculture on environment. Therefore,  I consider the conducted of the research described in the paper as very important.

The use of game theory to study the complex interactions between individual market stakeholders is undoubtedly a good proposition, especially that 4 different groups of the most important stakeholders were taken into account.

The assumptions are clearly defined. Of course, it is possible to try to suggest some bigger or lesser modification of their, but their presentation in the following points is consistent.

Perhaps it would be better to change the title of the work to: “Assessment of the problems related to the placing organic plant protection products  in rural region based on models study

In the abstract, the phrase "We have constructed ..." should be changed to "It was constructed ..." (line 11). In a few other places of the paper the personal form needs to be changed to impersonal as well.

Reviewer 3 Report

Notes are in the file.

Author Response

Thank you for your significant reminding.
